# Multimorbidity Patterns and Functioning Associations Among Adults in a Local South African Setting: A Cross-Sectional Study

**DOI:** 10.3390/ijerph22050780

**Published:** 2025-05-14

**Authors:** Karina Berner, Diribsa Tsegaye Bedada, Hans Strijdom, Ingrid Webster, Quinette Louw

**Affiliations:** 1Division of Physiotherapy, Department of Health and Rehabilitation Sciences, Faculty of Medicine and Health Sciences, Stellenbosch University, Cape Town 8000, South Africa; qalouw@sun.ac.za; 2Department of Statistics and Actuarial Science, University of Waterloo, P.O. Box 200, University Avenue West, Waterloo, ON N2L 3G1, Canada; diribsa2@gmail.com; 3Centre for Cardiometabolic Research in Africa, Division of Medical Physiology, Faculty of Medicine and Health Sciences, Stellenbosch University, Cape Town 8000, South Africa; jgstr@sun.ac.za (H.S.); iwebster@sun.ac.za (I.W.)

**Keywords:** activities of daily living, exercise tolerance, developing countries, functional status, hand strength, multimorbidity, primary healthcare, South Africa

## Abstract

Multimorbidity poses significant challenges for resource-constrained healthcare systems, particularly in low and middle income countries where specific combinations of chronic conditions may differentially impact function. This cross-sectional study examined multimorbidity patterns and associations with functioning among 165 adults attending semi-rural primary healthcare facilities in South Africa. Participants completed performance-based measures (handgrip strength, five-times sit-to-stand test, step test and exercise prescription tool [STEP] maximum oxygen consumption) and self-reported function (12-item WHODAS 2.0). Exploratory factor analysis identified three multimorbidity patterns: HIV-hypercholesterolaemia-obesity (Pattern 1), hypertension-anaemia-lung disease (Pattern 2), and stroke-heart disease-hypercholesterolaemia (Pattern 3). Pattern 1 was associated with reduced aerobic capacity (β = −6.41, 95% CI: −9.45, −3.36) and grip strength (β = −0.11, 95% CI: −0.14, −0.07). Pattern 2 showed associations with mild (β = 1.12, 95% CI: 0.28, 1.97) and moderate (β = 1.48, 95% CI: 0.53, 2.43) self-reported functional problems and reduced grip strength (β = −0.05, 95% CI: −0.09, −0.003). Pattern 3 was associated with all self-reported impairment levels, with the strongest association for severe impairment (β = 2.16, 95% CI: 0.32, 4.01). These findings highlight the convergence of infectious and non-communicable diseases in this setting. Simple clinical measures like grip strength and self-reported function may hold potential as screening or monitoring tools in the presence of disease patterns, warranting further research.

## 1. Introduction

Multimorbidity is becoming increasingly prevalent globally [1]. This trend is particularly concerning in low and middle income countries (LMICs), where approximately one-fifth of adults have multimorbidity, with rates potentially surpassing those in high-income countries (HICs) [2]. Multimorbidity poses significant challenges for already resource-constrained LMIC healthcare systems [3]. The poor health outcomes associated with having multiple health conditions include functional decline, disability, and reduced quality of life [4]; however, emerging evidence suggests that specific combinations of chronic conditions may have differential impacts on function [5]. This nuanced understanding of disease clustering and functional sequelae is crucial for developing targeted and optimal prevention and management.

Understanding which diseases tend to co-occur can provide insights into common underlying mechanisms and their collective impact on patient outcomes [6]. Studies have typically relied on basic or weighted disease counts, which overlook the diverse combinations of health issues and potential shared underlying mechanisms or risk factors. Subsequently, researchers are increasingly employing data-driven methods, like cluster analysis and exploratory factor analysis, to uncover multimorbidity patterns [7]. Systematic reviews, although primarily focusing on HICs, have identified relatively consistent patterns, including cardiovascular and metabolic diseases, mental health, and musculoskeletal clusters [8,9]. However, the LMIC multimorbidity landscape presents unique characteristics and challenges. Additional clusters involving long-term infectious diseases, like HIV, have recently been reported in these contexts [10,11]. The functional impacts of these patterns may differ significantly in LMICs due to several factors, including earlier multimorbidity onset, different disease combinations, and unique healthcare and socioeconomic challenges [12,13].

Studies on multimorbidity patterns and functioning associations in LMICs have reported that chronic lung disease, tuberculosis, and mental–sensory combinations are associated with the worst outcomes [10,14]. A recent scoping review [15] suggests that while certain functioning problems (mobility, instrumental activities of daily living, self-care, and bowel/bladder function) are consistently associated with various multimorbidity patterns in LMICs, the “cardiovascular and metabolic diseases” domain appears to be linked with the broadest range of functional issues. However, the review highlighted significant gaps in current knowledge, including a limited geographical scope (89% of studies from Asia), a focus on older adults, and methodological heterogeneity. These limitations are particularly concerning given that multimorbidity in LMICs often affects younger populations compared with HICs [12], potentially impacting economic productivity and increasing the societal burden of chronic diseases. Comprehensive evidence on multimorbidity patterns and their functional impacts across diverse settings and age groups remains lacking.

Despite the growing recognition of multimorbidity as a major public health challenge, most LMIC healthcare systems remain structured around single-disease care models. This approach is often inadequate for managing the complex needs of individuals with multiple chronic conditions. There is an urgent need for integrated, person-centred approaches to care that consider the full spectrum of an individual’s health needs and functional status. As LMICs strive to achieve universal health coverage and shift towards primary care-led chronic disease management, understanding multimorbidity patterns and any functioning implications is timely. This knowledge can inform the development of context-appropriate clinical guidelines, health policies, and service delivery models that address the complex needs of individuals with multimorbidity in resource-limited settings. This study aimed to describe multimorbidity patterns and investigate their relationships with functioning measures among adults in a local primary healthcare setting.

## 2. Materials and Methods

### 2.1. Study Design, Setting, and Recruitment

This cross-sectional study was conducted in semi-rural primary care community health centres (CHCs) in the Cape Winelands District, South Africa. Between January 2020 and February 2024, potential participants were recruited from two sequential studies accessing the same study cohort and led by the Centre for Cardiometabolic Research in Africa, Division of Medical Physiology, Stellenbosch University: the EndoAfrica longitudinal cohort [16] and the ongoing MITO-SAKen cross-sectional study. Both studies examined cardiovascular health outcomes in adults with and without HIV in the Western Cape Province, South Africa. The flow of participant recruitment and inclusion is shown in Figure 1.

The study population consisted of adult men and women with and without HIV attending these primary healthcare facilities. Eligibility criteria aligned with the parent studies and included being 18 years or older and providing written informed consent. Women who were pregnant or less than three months postpartum were excluded, as were individuals with active tuberculosis or those who were acutely ill. For the current analysis, participants additionally had to have multimorbidity, i.e., report or signs of at least two chronic conditions, as per Appendix A. Dedicated research nurses recruited participants from the parent studies, and potentially eligible participants enrolled in these cohorts were simultaneously and consecutively invited to participate in the present study.

### 2.2. Sample Size Considerations

Sample size determination for exploratory factor analysis (EFA) is complex and depends on multiple factors [17]. With 15 chronic conditions considered in the multimorbidity definition for the purposes of this study, a minimum sample size of *n* = 150 was targeted, based on common recommendations [18]. The final sample of *n* = 165 exceeded this target, potentially providing a reasonable basis for stable factor solutions. However, the adequacy of sample size ultimately depends on the strength of factor loadings and communalities obtained in the analysis [19]. The Kaiser–Meyer–Olkin (KMO) measure of sampling adequacy yielded a value of 0.56 for the sample, exceeding the minimum threshold of 0.50 but not the more ideal threshold of 0.60, indicating only moderate adequacy [17].

### 2.3. Health Conditions and Multimorbidity

Multimorbidity was defined as the co-occurrence of at least two conditions from a pre-defined list of 15 health conditions. These conditions were selected based on their common inclusion in multimorbidity studies and data availability from the parent studies. Detailed criteria for defining each health condition are provided in Appendix A. Multimorbid patterns were described as the co-occurrence of at least two conditions from the list of 15, considering only conditions with a prevalence of ≥ 3% in the sample to avoid spurious associations and obtain epidemiologically coherent patterns [14,15].

Clinical and blood and urine biochemical data, extracted from the parent studies, were used to identify signs of selected health conditions. These included total cholesterol, high-density lipoprotein cholesterol (HDL), low-density lipoprotein cholesterol (LDL), triglycerides (markers of dyslipidaemia), fasting glucose, glycated haemoglobin (HbA1c) (markers of diabetes mellitus), haemoglobin (marker of anaemia), estimated glomerular filtration rate (eGFR), albumin to creatinine ratio (ACR) (markers of renal impairment), gamma-glutamyl transpeptidase (GGT), alanine aminotransferase (ALT), aspartate aminotransferase (AST), and alkaline phosphatase (ALP) (markers of liver impairment). As part of the parent studies, a research nurse collected the blood specimens (approximately 40 mL) and urine samples. The samples were transported to the Division of Medical Physiology, Stellenbosch University, Tygerberg Campus, for standard biochemical analyses at the NHLS, Tygerberg.

Self-reported conditions were ascertained by asking participants: “Have you ever been told by a doctor or healthcare worker that you have any of the following, or do you take prescription medication for any of the following…”. Conditions assessed solely by self-reporting included arthritis (osteo- or rheumatoid), chronic lung disease (asthma, chronic bronchitis, emphysema, chronic obstructive pulmonary disease [COPD], or a persistent cough with phlegm daily for at least three months of the year for at least two consecutive years), and cataracts. Depression was also identified through self-reported diagnosis, along with current use of prescription medication, or Centre for Epidemiologic Studies Short Depression Scale 10 (CES-D-10) scores. The CES-D-10, validated in South African English, Afrikaans, and isiXhosa [20], used cut-off scores of ≥11 for Afrikaans, ≥10 for English, and ≥13 for isiXhosa participants [20].

Several conditions were assessed using both self-reported and clinical measurements: hypertension (self-reported diagnosis and use of antihypertensive medication, or blood pressure ≥ 130/85 mmHg), diabetes (self-reported diagnosis and use of medication, or fasting glucose ≥ 7.0 mmol/L or HbA1c ≥ 6.5%), kidney impairment (self-reported diagnosis and medication use, or eGFR < 60 mL/min/1.73 m^2^ or ACR ≥ 3.0 mg/mmol creatinine), liver impairment (self-reported diagnosis, or at least two of ALT > 35 U/L, AST > 35 U/L, ALP > 98 U/L, GGT > 40 U/L), anaemia (self-reported diagnosis or haemoglobin < 12.0 g/dL), heart disease (self-reported history of heart attack or diagnosis with current use of specific cardiac medications, particularly nitrates for angina or antiplatelet drugs), and high blood cholesterol (self-reported diagnosis, use of cholesterol-lowering medication, or total cholesterol ≥ 5.17 mmol/L or LDL cholesterol > 3.0 mmol/L).

HIV status was confirmed through medical records for individuals with known HIV. For those with unknown status, rapid HIV testing was performed with pre- and post-test counselling. All participants voluntarily consented to HIV testing and status disclosure as a study requirement. Signs of obesity were identified by body mass index (BMI) ≥ 30 kg/m^2^ or waist circumference ≥ 94 cm (men)/≥80 cm (women).

### 2.4. Functioning Outcomes

Performance-based functioning outcomes included handgrip strength (HGS), lower limb function measured using the five-times sit-to-stand test (5STS), and the STEP prediction tool of VO_2_max. Self-reported functional impairment was assessed using the 12-item version of the World Health Organisation Disability Assessment Schedule (WHODAS 2.0).

HGS has been shown to be both a useful marker of multimorbidity patterns and a valid predictor of functional impairment in populations with multiple chronic conditions [21,22] and was assessed using a calibrated hydraulic hand dynamometer (Model SH5001; SAEHAN Corporation, Masan, Gyeongsangnam-do, Republic of Korea). Following standardised procedures, participants performed three maximal voluntary contractions with each hand in alternating order, with 30 s of rest between trials on the same hand. Testing was conducted by sitting on a standard height chair, with the feet on the ground, shoulder adducted, elbow flexed to 90 degrees, and forearm in neutral position. The average value of the three attempts with the dominant hand was recorded in kilograms and normalised to body mass (kg grip strength/kg body mass) to account for the influence of body size on absolute strength values [23].

Lower limb function was evaluated using the five-times sit-to-stand test (5STS) [24], which assessed muscle strength, power, dynamic balance, and functional mobility. Participants performed the test on a standard chair with a firm seat and stable backrest. Starting from a seated position with feet placed naturally and knees approximating 90 degrees [25], participants stood up five times as quickly as possible, keeping their arms folded across their chest. The test concluded when participants reached the final standing position. After practising two repetitions, participants completed two timed trials, with the fastest time (measured to 0.01 s) being selected for analysis.

Functional aerobic capacity was assessed using the step test and exercise prescription (STEP™) tool protocol [26]. Participants completed 20 cycles of stepping up and down a standardised two-step platform (20 cm per step) at a self-selected pace [27]. After one or two practice cycles, the test was performed once, with the completion time, post-test heart rate, body weight, age, and sex recorded for the STEP tool VO_2_max prediction equation.

Self-reported activity limitations were assessed using the 12-item World Health Organization Disability Assessment Schedule (WHODAS 2.0) in English, Afrikaans, or isiXhosa as per participant preference. The WHODAS 2.0 is a validated measure based on the International Classification of Functioning, Disability, and Health framework and evaluates functioning across six domains: cognition, mobility, self-care, relationships, life activities, and social participation [28]. Each item is scored on a five-point Likert scale ranging from ‘no difficulty’ to ‘extreme/cannot do’. Total scores were converted to percentages and categorised as: no disability (0–4%), mild (5–24%), moderate (25–49%), severe (50–95%), or extreme disability (96–100%).

### 2.5. Sociodemographic and Lifestyle-Related Variables

Additional data extracted from parent studies to include in the current analysis included sociodemographic characteristics (age, sex, education level, monthly household income, employment status), lifestyle factors (smoking status: yes/no answer to “ever smoked” and alcohol consumption over the past 12 months), and clinical measures (BMI, sphygmomanometry).

### 2.6. Procedures

Potentially eligible participants from the existing study cohorts were consecutively invited by a research nurse to enrol for the current study assessments. After obtaining informed consent to partake in both study components, participants completed a personal interview and comprehensive health questionnaire administered by the research nurse at the relevant clinic. Standard anthropometric measurements and a cardiovascular system-focused physical examination were performed by the nurse in a private room. Blood and urine samples were collected for biochemical and biomarker analyses to confirm chronic comorbidities. HIV status was determined or confirmed through rapid testing, with pre- and post-test counselling provided by the research nurse.

Physical function assessments were conducted in a second, private, quiet room by a qualified physiotherapist (K.B. and a research assistant), according to the standardised protocols described earlier. Participants first completed the 12-item WHODAS 2.0 and CES-D10, followed by the functional performance tests in randomised order.

### 2.7. Statistical Analysis

Descriptive statistics were calculated for all variables. For continuous data, normality was assessed using the Shapiro–Wilk test and means and standard deviations (SDs) were reported where distributions were normal, while medians and ranges were used for non-symmetrical distributions. Categorical data were presented as frequencies and percentages.

#### 2.7.1. Exploratory Factor Analysis (Pattern Analysis)

An exploratory factor analysis (EFA) was applied to determine associations between diseases that demonstrate multimorbidity patterns. Multimorbidity was defined as the presence of two or more chronic conditions from the predefined list of 15 conditions. Additionally, from this list, only conditions with a prevalence ≥ 3% were included in the factor analysis to increase epidemiological relevance [14]. EFA was chosen as it allows for variables (diseases) to be included in more than one factor (pattern), which is particularly useful for considering pathophysiological relationships in multimorbidity research [29].

The factor analysis employed a correlation matrix to identify which diagnostic variables (diseases) constituted each factor (multimorbid pattern). Given that study variables were dichotomous (1/0 indicating presence/absence of each disease), tetrachoric correlations were used to calculate the correlation matrix between diagnoses. The analysis proceeded on the assumption that dichotomous diagnoses have underlying continuous characteristics, specifically, that the included chronic conditions progress gradually and are diagnosed when they cross a particular threshold [30]. The patterns of multimorbidity (i.e., chronic diseases commonly occurring together) were interpreted from the factors emerging from the EFA. The association between each disease and its corresponding disease pattern was represented by factor scores (values ranging from −1 to 1). The Kaiser–Meyer–Olkin (KMO) measure was applied to assess the sample’s adequacy for factor analysis. The initial solution was extracted using the principal component factor method. To determine the optimal number of factors for the final solution, a scree plot displaying eigenvalues of the correlation matrix in descending order was utilised. The number of extracted factors corresponded to the eigenvalue’s sequence number that created the curve’s inflection point and exceeded 1 (Kaiser criterion).

Factor patterns were identified through an analysis of factor loadings. An oblique rotation (Oblimin) was applied to elucidate the factor pattern and enhance interpretation of factor characteristics, based on the assumption that factors may correlate with one another. Diseases constituting each multimorbidity pattern were determined by factor loadings with absolute values ≥ 0.25, which were deemed significant [31].

The author team evaluated the clinical relevance of the final EFA solution by comparing identified patterns with the existing literature and assessing the plausibility of disease co-occurrences based on their collective research and clinical expertise.

#### 2.7.2. Associations with Functioning Outcomes

Several regression analyses were conducted. Factor scores were predicted for each participant, representing their alignment with each identified multimorbidity pattern. Multiple linear regression models were used for continuous outcomes: VO_2_max, normalised HGS, and chair rise time. Multinomial logistic regression was used for the categorical outcome, i.e., WHODAS-12 activity limitation class. All models included the factor scores as predictors, representing the three identified multimorbidity patterns. Age, sex, and smoking status were included as covariates in most models to adjust for potential confounding effects [15]. For the VO_2_max outcome, only smoking status was included as a covariate due to age and sex already being incorporated in the STEP VO_2_max calculation.

All statistical analyses were performed using Stata Version 18 (StataCorp., College Station, TX, USA). Statistical significance was set at *p* < 0.05 for all analyses.

### 2.8. Ethical Considerations

The study was approved by Stellenbosch University’s Health Research Ethics Committee (HREC) as an amendment to the existing projects (N19/02/029 and N22/11/137). All participants provided written informed consent.

## 3. Results

### 3.1. Participant Characteristics

Table 1 presents an overview of the study population’s (*n* = 165) sociodemographic and health characteristics, subjective activity limitations, and physical performance metrics. The participants’ mean age was 41.58 years, with women comprising two-thirds of the sample (66.10%). While men and women exhibited similarities across most parameters, men demonstrated significantly higher STEP VO_2_max (*p* = 0.000). The median WHODAS-12 percentage score was 8.33% (range 0.00–79.17%), with 68.9% of participants reporting any level of activity limitation. Physical performance measures revealed a mean non-normalised HGS of 27.78 kg (SD 7.62 kg), a median normalised HGS of 0.43 (SD 0.15), a mean chair rise time of 9.17 s (SD 2.00), and a mean STEP VO_2_max of 42.4 (SD 17.1).

The frequency of occurrence for the 15 health conditions is illustrated in Figure 2. Conditions exhibiting a prevalence of ≥3% were chosen for further analysis [14], resulting in a final selection of 13 conditions for the subsequent factor analysis (cancer and cataracts, each with a 1% prevalence, were omitted).

### 3.2. Multimorbidity Patterns

After examining the sampling adequacy of the variables for factor analysis, the overall KMO score was initially found to be 0.49, which is well below the recommended threshold of 0.60. To improve this, 5 of the remaining 13 diseases were removed from the analysis (arthritis, depression, diabetes, liver impairment, and renal impairment) as they were highly correlated with the other variables in the dataset and contributed to the low KMO score. After excluding these, the KMO improved to 0.56. Although still below 0.60, this was deemed a more suitable starting point. The scree plot indicated that the number of factors extracted was three (Figure 3).

The first factor (Pattern 1) comprised HIV, cholesterol, and obesity. The second factor (Pattern 2) consisted of hypertension, anaemia, and lung disease; and the third factor (Pattern 3) of stroke, heart disease, and cholesterol (Table 2).

### 3.3. Associations Between Multimorbidity Patterns and Functioning

Associations between multimorbidity patterns and physical performance are shown in Table 3. Pattern 1 showed significant negative associations with STEP VO_2_max (β = −6.41, 95% CI: −9.45, −3.36, *p* < 0.001) and normalised HGS (β = −0.11, 95% CI: −0.14, −0.07, *p* < 0.001). Pattern 2 was significantly associated with mild (WHODAS-12 5–24%; β = 1.12, 95% CI: 0.28, 1.97, *p* < 0.01) and moderate (WHODAS-12 25–49%; β = 1.48, 95% CI: 0.53, 2.43, *p* < 0.01) self-reported functional impairment and reduced normalised HGS (β = −0.05, 95% CI: −0.09, −0.003, *p* < 0.05). Pattern 3 demonstrated significant associations with all self-reported impairment levels, with the strongest association observed for severe impairment (WHODAS-12 96–100%; β = 2.16, 95% CI: 0.32, 4.01, *p* < 0.05). No significant associations were observed between multimorbidity patterns and chair rise performance (Pattern 1: β = −0.40 s, Pattern 2: β = −0.18 s, Pattern 3: β = −0.60 s); however, the overall model fit was poor (R^2^ = 0.026, *p* = 0.784). These changes were smaller than the minimal clinically important difference values for the five-times sit-to-stand test (2.5 s) [32].

## 4. Discussion

This study aimed to identify multimorbidity patterns and examine their associations with functioning among adults attending primary healthcare facilities in a local semi-rural South African setting. Factor analysis identified three patterns of co-occurring conditions: (1) HIV, hypercholesterolaemia, and obesity; (2) hypertension, anaemia, and chronic lung disease; and (3) stroke, heart disease, and hypercholesterolaemia. These patterns showed differential associations with functioning outcomes. The first pattern was associated with reduced aerobic capacity and HGS, the second showed associations with self-reported limitations and reduced HGS, while the third was associated with self-reported limitations and reduced aerobic capacity.

While statistical methods can identify disease patterns, interpreting these patterns requires careful consideration. The risk factor combinations identified as distinct patterns via the factor analysis in this study do not necessarily reflect direct pathophysiological relationships or distinct clinical entities. There may often be significant pathophysiological overlap between conditions that appear in different statistical clusters—for example, the cardiometabolic continuum means that many conditions traditionally categorised as either “metabolic” or “cardiovascular” share common underlying mechanisms [33]. Similarly, although chronic lung disease, heart disease, and stroke share substantial biological pathways such as systemic inflammation, endothelial dysfunction, and common risk factors like smoking [34], these conditions did not cluster together statistically in our analysis. It may be that the EFA identification of patterns of statistical covariation was influenced by disease prevalence, severity, and the characteristics of our sample rather than purely underlying biological mechanisms [6,30]. Additionally, the moderate KMO value in the factor analysis suggests some limitations in sampling adequacy, although the final three-factor solution demonstrated statistical coherence.

The co-occurrence of HIV with hypercholesterolaemia and obesity in one pattern demonstrates the convergence of infectious and non-communicable diseases that characterises multiple disease burdens observed in many LMICs and aligns with recent analyses in the South African setting [11]. This clustering of HIV with cardiometabolic conditions has been increasingly documented, particularly among ART (antiretroviral therapy)-experienced individuals, where HIV has been reported to co-occur with hypertension, dyslipidaemia, diabetes, cancer, and cardiovascular disease, including findings from the EndoAfrica study cohort [35,36,37]. The identification of hypertension-anaemia-lung disease and cardiovascular patterns parallels findings in both local and international literature—similar groupings were identified in a South African latent class analysis [11], while systematic reviews have relatively consistently identified “cardiometabolic” (including hypertension, diabetes, obesity), “respiratory” (including chronic lung diseases), and “cardiovascular” (including stroke and heart disease) as common patterns globally [8,9].

The low cancer prevalence in our sample aligns with findings from other South African studies [36,38]. Although speculative, this observation could be attributed to several factors: our sampling approach (relatively young, predominantly female and non-randomised), potential under-diagnosis and/or under-reporting (a known issue in South African settings) [39,40,41], and the epidemiological profile of multimorbidity in this context. In South African HIV cohorts, multimorbidity appears primarily characterised by cardiometabolic conditions and infectious diseases rather than malignancies [36].

The emergence of these patterns in a relatively young sample (mean age 41.6 years) reinforces evidence that multimorbidity may manifest early in LMICs [12], likely reflecting complex interactions between biological, socioeconomic, and healthcare system factors. This is supported in a recent systematic review [42], which found significant associations between multimorbidity patterns and socioeconomic status, sex, and health behaviours, highlighting the need for further research on these complex interactions. Furthermore, the seemingly consistent identification of certain disease combinations across various studies using different statistical methodologies suggests that such patterns, while statistically derived, may reflect meaningful disease clustering influenced by both pathophysiological mechanisms and broader social determinants of health. Gaining further understanding into these complexities will have important implications for both clinical practice and health system planning in South Africa’s resource-constrained context.

The functioning performance characteristics of the sample revealed that men demonstrated significantly higher VO_2_max values (as expected) but similar absolute grip strength compared with women—an unexpected finding. These sex-based differences warrant further investigation, particularly given previous research showing sex-based disparities in multimorbidity prevalence and healthcare utilisation [11,42]. The overall relatively high mean VO_2_max values should be interpreted cautiously, as the STEP™ protocol has been reported to systematically overestimate VO_2_max (reportedly by 12% or 6.4 mL/kg/min) compared with direct measures [26,43]. The small sex difference in absolute HGS—much lower in men than expected according to population norms—could suggest differential relationships between HGS and multimorbidity patterns across sexes [21,44]. However, evidence regarding such sex-specific associations remains mixed in the literature.

Differential associations were observed between multimorbidity patterns and functioning measures. Evidence suggests that co-occurring diseases may influence functional capacity through multiple, often interconnected pathways [45]. HGS has been reported to serve as an indicator of functionality across various physiological systems, and it may be valuable to monitor as an indicator of overall health status and risk of developing multiple chronic conditions [21]. This role of HGS as a potential marker of multimorbidity was evident in our findings, showing associations with two of the identified patterns. The absence of associations with chair rise performance is in contrast to a recent report that various multimorbidity patterns—including those involving stroke and cardiovascular disease—are associated with impaired chair rise performance [46]. However, the relatively younger average age of our sample (41.58 years versus 69.94 years in [46]) may support the role of age as a moderator of the relationship between disease combinations and function [47], with younger populations potentially maintaining sufficient functional reserve to compensate for certain disease combinations. Moreover, the differential associations observed in our study may feasibly reflect varying levels of disease severity and additional factors such as polypharmacy, treatment burden, psychosocial factors, and social network characteristics [45,48], underscoring the need for longitudinal research investigating these complex relationships.

Nevertheless, the observed associations between multimorbidity patterns and simple, quick, and inexpensive clinical assessments of functioning suggest potential clinical applications. These tools could potentially serve as accessible screening or monitoring measures in resource-constrained primary healthcare settings, where comprehensive clinical assessments are not always feasible. For example, the identification of high-risk multimorbidity patterns through these measures could facilitate early intervention and promote better health outcomes.

Study strengths include the use of consecutive sampling and standardised assessment protocols by a qualified physiotherapist, which helped ensure systematic data collection. The combination of objective measures and self-reported outcomes provided complementary perspectives on functional status. Offering questionnaires in multiple local languages enhanced accessibility and representation. However, several limitations must also be acknowledged when interpreting the findings. The moderate KMO value suggests some limitations in sampling adequacy, and the cross-sectional design precludes causal inference. The conditions included in the pattern analysis were limited pragmatically by what was available from the data sources. Furthermore, the primary care recruitment setting may have influenced the observed disease co-occurrences, potentially overestimating disease prevalence compared with the general population. Given that participants were recruited from larger studies focusing on cardiovascular health in people with and without HIV, the generalisability of findings is likely limited to similar semi-rural primary healthcare settings, with potentially higher HIV prevalence than the general population, and may not extend to broader South African contexts, older populations, or different healthcare settings.

Future longitudinal research is needed to better understand how different disease combinations influence functional trajectories over time in adults with multimorbidity. Such studies should consider disease severity, treatment burden, and social determinants when examining these relationships. Investigation of how rehabilitation interventions might be optimised for different multimorbidity patterns could provide valuable insights for clinical practice. Additionally, research exploring the mechanisms through which multiple conditions interact to affect different functional domains would enhance understanding of assessment and intervention priorities. Studies examining how factors like polypharmacy and social support moderate the relationship between multimorbidity and function could help identify important targets for comprehensive care approaches.

## 5. Conclusions

This study provides novel evidence about multimorbidity patterns and their functioning associations in a relatively young local South African sample. Factor analysis identified three multimorbidity patterns: HIV, hypercholesterolaemia, and obesity; hypertension, anaemia, and chronic lung disease; and stroke, heart disease, and hypercholesterolaemia. These combinations showed differential functional associations, and the findings suggest that HGS, the WHODAS 2.0, and the STEP tool may warrant further investigation as simple, non-invasive tools in screening for or monitoring certain patterns of multimorbidity. Future longitudinal research is needed to better understand how different disease combinations influence functional trajectories over time (or vice versa), considering disease severity, treatment burden, and social determinants.

## Figures and Tables

**Figure 1 ijerph-22-00780-f001:**
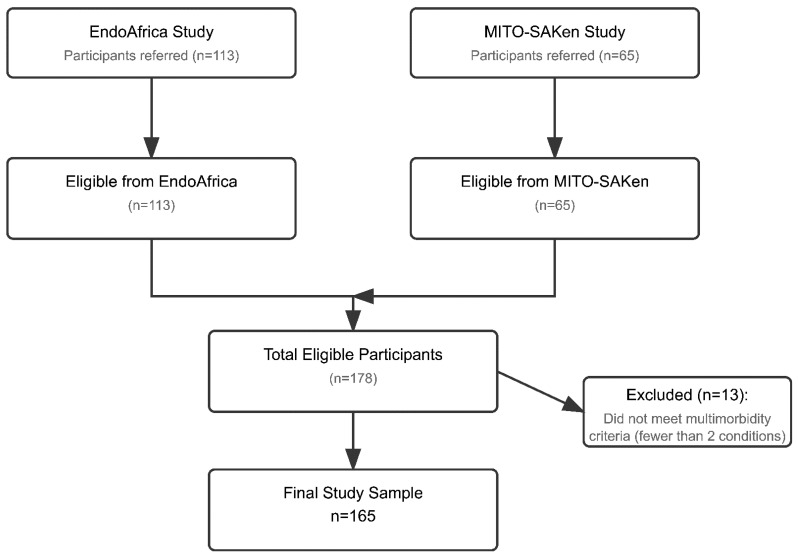
Flow diagram of participant recruitment and inclusion.

**Figure 2 ijerph-22-00780-f002:**
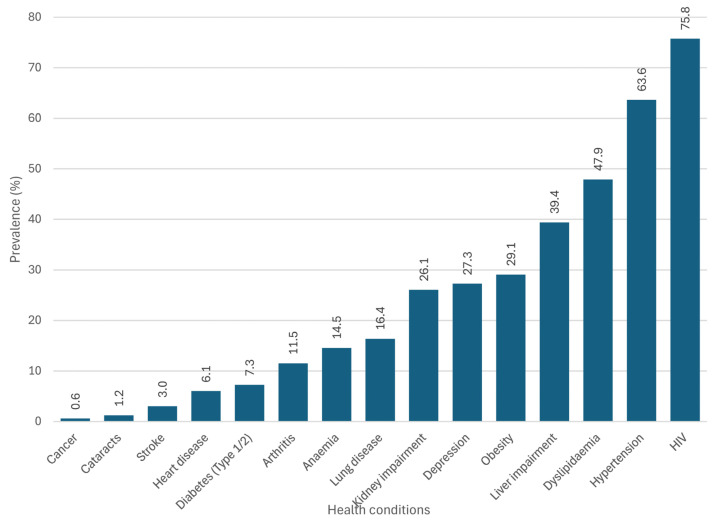
Prevalence of the 15 individual health conditions in the sample (*n* = 165).

**Figure 3 ijerph-22-00780-f003:**
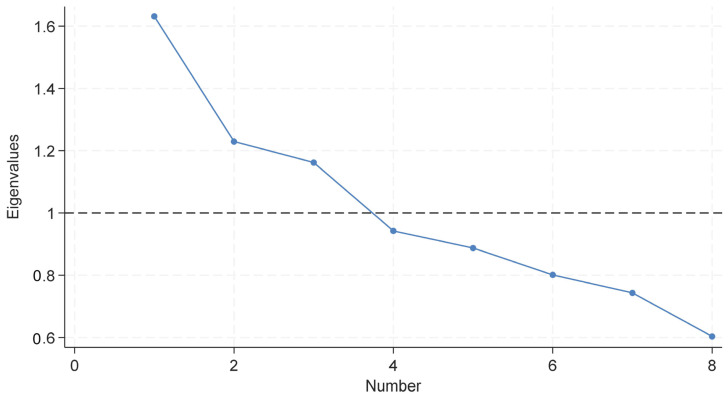
Scree plot of eigenvalues after factor analysis (*n* = 165). The x-axis indicates the number of factors. The dashed horizontal line represents the Kaiser criterion (eigenvalue = 1), used to guide factor retention. Although the plot did not show a clear elbow, three factors were retained based on the Kaiser criterion (visualised above the dashed line).

**Table 1 ijerph-22-00780-t001:** Sociodemographic, lifestyle, health, and functioning characteristics of participants (*n* = 165).

Characteristic	Total	Women (*n* = 109)	Men (*n* = 56)	*p*-Value
Age in years, mean (SD)	41.58 (9.99)	40.71 (10.06)	43.27 (9.71)	0.119
Education level, *n* (%)				
None	11 (6.67%)	8 (7.34%)	3 (5.36%)	
Primary school	67 (40.61%)	48 (44.04%)	19 (33.93%)	
High school	72 (43.64%)	45 (41.28%)	27 (48.21%)	
ABET (adult basic education training)	1 (0.61%)	0 (0.00%)	1 (0.92%)	0.451
College/University/Other tertiary institution	14 (8.48%)	7 (6.42%)	7 (12.50%)	
Unemployed, *n* (%)	93 (56.36%)	64 (58.72%)	29 (51.79%)	0.395
Monthly household income, *n* (%)				
<1000 ZAR	32 (19.39%)	20 (18.35%)	12 (21.43%)	
≥1000 ZAR to <5000 ZAR	88 (53.33%)	64 (58.71%)	24 (42.86%)	0.126
≥5000 ZAR	45 (27.27%)	25 (22.94%)	20 (35.71%)	
Current or former smoker, *n* (%)	104 (63.03%)	68 (62.39%)	36 (64.29%)	0.811
Alcohol consumption in last 12 months, *n* (%)	86 (52.12%)	58 (50.00%)	28 (53.21%)	0.696
Functioning				
WHODAS-12 percentage score, median (range)	8.33 (0.00–79.17)	10.42 (0.00–66.67)	6.25 (0.00–79.16)	0.213
No limitation (0–4%), *n* (%)	51 (31.10%)	32 (29.36%)	19 (34.55%)	
Mild limitations (5–24%), *n* (%)	72 (43.90%)	48 (44.04%)	24 (43.64%)	
Moderate (25–49%), *n* (%)	23 (20.12%)	33 (21.10%)	10 (18.18%)	0.865
Severe (50–95%), *n* (%)	8 (4.88%)	6 (5.50%)	2 (3.64%)	
Extreme/Cannot do (96–100%), *n* (%)	0 (0.00%)	0 (0.00%)	0 (0.00%)	
Chair rise time in seconds, mean (SD)	9.17 (2.88)	9.25 (3.19)	9.02 (2.20)	0.650
STEP VO_2_max, mean (SD)	42.4 (17.1)	39.1 (16.9)	49.1 (15.7)	**0.000**
Handgrip strength in kilograms, mean (SD)	27.78 (7.62)	27.57 (7.82)	28.18 (7.26)	0.318
Normalised handgrip strength, mean (SD)	0.43 (0.15)	0.44 (0.15)	0.44 (0.14)	0.495

Note: SD—Standard deviation; STEP—step test and exercise prescription tool; WHODAS—World Health Organisation Disability Assessment Schedule; ZAR–South African Rand. Categorical variables tested using chi-squared test, continuous variables tested using independent *t*-test or the Wilcoxon rank sum test. Bold values are used to represent where *p* < 0.05.

**Table 2 ijerph-22-00780-t002:** Rotated factor loadings of health conditions across three multimorbidity patterns.

Health Condition	Factor		
	HIV Pattern (Pattern 1)	Hypertension Pattern (Pattern 2)	Cardiovascular Pattern (Pattern 3)
Stroke	0.0109	−0.0027	**0.3019**
Obesity	**0.3518**	0.1562	−0.1075
Lung disease	−0.0553	**0.2651**	−0.1381
Heart disease	0.2295	−0.0034	**0.2845**
HIV	**−0.5211**	0.0840	0.0130
Hypertension	0.1866	**−0.3855**	0.0268
Hypercholesteraemia	**0.3844**	−0.1044	**0.2503**
Anaemia	−0.0834	**0.3178**	0.0688

Note: Kaiser–Meyer–Olkin value = 0.5586; factor loadings indicate the strength of association between each health condition and each factor (multimorbidity pattern), with factor loadings of ≥±0.25 in bold.

**Table 3 ijerph-22-00780-t003:** Associations between multimorbidity patterns and functional outcomes.

Pattern	Unadjusted Analysis				Adjusted Analysis			
	Step Test VO_2_max	Normalised Handgrip Strength	Chair Rise Time	WHODAS-12 Impairment Level	Step Test VO_2_max ^1^	Normalised Handgrip Strength ^2^	Chair Rise Time ^2,3^	WHODAS-12 Impairment Level ^4^
Pattern 1 (HIV-cholesterol-obesity)								
β (95% CI)	**−6.60 (−9.59, −3.60) *****	**−0.10 (−0.14, −0.07) *****	−0.30 (−1.06, 0.45)	Mild: −0.11 (−0.68, 0.46)	**−6.41 (−9.45, −3.36) *****	**−0.11 (−0.14, −0.07) *****	−0.40 (−1.17, 0.38)	Mild: 0.05 (−0.55, 0.64)
				Moderate: −0.67 (−1.46, 0.12)				Moderate: −0.54 (−1.35, 0.26)
				Severe: −0.95 (−2.48, 0.59)				Severe: −1.75 (−3.68, 0.19) †
Pattern 2 (hypertension-anaemia-lung disease)								
β (95% CI)	−2.36 (−5.95, 1.22)	−0.04 (−0.08, 0.005) †	−0.13 (−1.04, 0.78)	**Mild: 0.85 (0.07, 1.64) ***	−2.41 (−6.09, 1.28)	**−0.05 (−0.09, −0.003) ***	−0.18 (−1.12, 0.76)	**Mild: 1.12 (0.28, 1.97) ****
				**Moderate: 1.30 (0.41, 1.48) ***				**Moderate: 1.48 (0.53, 2.43) ****
				**Severe: 1.69 (0.54, 2. 84) ***				Severe: 1.50 (0.00, 3.01) †
Pattern 3 (stroke-heart disease-cholesterol)								
β (95% CI)	−0.15 (−4.13, 3.83)	0.04 (−0.004, 0.09) †	−0.54 (−1.55, 0.47)	Mild: −0.95 (−2.48, 0.59)	−0.15 (−4.14, 3.85)	0.04 (−0.01, 0.09) †	−0.60 (−1.63, 0.42)	**Mild: 1.14 (0.06, 2.21) ***
				Moderate: 1.27 (−0.16, 2.70)				**Moderate: 1.66 (0.47, 2.84) ****
				**Severe: 1.90 (0.39, 3.43) ***				**Severe: 2.16 (0.32, 4.01) ***
Model Statistics								
R^2^	0.118 ^5^	0.183 ^5^	0.014 ^5^	0.058 ^6^	0.124 ^5^	0.211 ^5^	0.026 ^5^	0.117 ^6^

Notes: ^1^ Adjusted for smoking status. ^2^ Adjusted for age, gender, and smoking status. ^3^ Overall model not statistically significant (F-test *p* = 0.784). ^4^ Adjusted for age, gender, and smoking status; WHODAS-12 categories: no disability (reference), mild, moderate, and severe impairment. ^5^ R^2^ from linear regression. ^6^ Pseudo R^2^ from multinomial logistic regression. Bold font indicates statistically significant results at *p* < 0.05. † *p* < 0.10, * *p* < 0.05, ** *p* < 0.01, *** *p* < 0.001.

## Data Availability

The datasets presented in this article are not readily available because the data are part of an ongoing study. Requests to access the datasets should be directed to the corresponding author.

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
