# Peer review of "Multimorbidity Patterns and Functioning Associations Among Adults in a Local South African Setting: A Cross-Sectional Study"

_ijerph, 2025, doi:10.3390/ijerph22050780_

Round 1
Reviewer 1 Report
Comments and Suggestions for Authors
The manuscript examines multimorbidity patterns and their association with functioning in a local South African cohort. Overall, it is clearly written, well-structured, and presents relevant findings. The discussion section effectively interprets the results and provides additional insights, and the authors have clearly acknowledged limitations such as sample size and potential population bias.
Below are a few comments and suggestions for the authors:
- The cohort consists entirely of individuals with multimorbidity (i.e., ≥ 2 conditions). Would it be possible to include a comparison group of healthy individuals with similar sociodemographic characteristics as a baseline? Including such a group could provide additional insights into the functional impacts of multimorbidity.
- For Table 1, in the row for “Monthly household income”, the category “≥1000 ZAR to < 5000 ZAR” has a misplaced parenthesis. Please revise to clearly reflect the percentage.
- On the x-axis of Figure 2, the health condition labeled as “depressive…” may be unclear to readers. Consider changing the label to “depression” to improve clarity.
- For “Pattern 3” in Table 2, “Heart disease” also appears as a component for Pattern 3 but is not bolded. To maintain consistency with how primary contributors are highlighted, please bold this entry as well.
- Under “Unadjusted analysis” in Table 3, the subscript notes (1, 2, 3, 4) indicating adjustment factors may be confusing, as it suggests the analysis may have been adjusted. If this section is indeed unadjusted, please consider removing or revising the notes to enhance clarity.
Reviewer 2 Report
Comments and Suggestions for Authors
It was surprising to me that the prevalence of cancer was so low. (Figure 2) Is this a sampling issue? It deserves a comment.
Lines 350-359. I appreciate the acknowledgement that the patterns identified by factor analysis don't necessarily reflect direct pathophysiological relationships; however, as you say, there may be significant pathophysiological overlap between conditions. I was particularly surprised at the lack of overlap between lung disease and heart disease and stroke. The pathophysiological overlap is clear and I think this needs more attention in the discussion.
Figure 3 (Scree Plot). Figure caption should identify the meaning of the dashed line (Kaiser criterion) and that the x-axis 'number' refers to the number of factors. The plot does not demonstrate a 'typical' or clear elbow shape. As this analysis technique may be new to many readers, it would be helpful to provide a description of how the interpretation was conducted.
Appendix A. This table could be more concise and therefore, easier for the reader to use. Much of the information in the column 'Description of data collection' is in the text or could easily be placed there. (blood and urine collection) In that case the table could just include cut-off values and specific information on blood pressure and HIV assessment. The text states that self-reported diagnosis is always an inclusion criterion so there is no need to put it in the 'Cut-off values/interpretation' column.
Line 182. Please provide a reference for the 5 time sit-to-stand test. The included reference (23) describes how to conduct the test.
Define ZAR (Table 1) and ART (Line 364). While ZAR is included in the list of abbreviations (page 17/25) it does not defined in the text the first time it is used as is the case with other abbreviations such as KMO. ART is not included in the list of abbreviations.
Reviewer 3 Report
Comments and Suggestions for Authors
South Africa has a large burden of HIV inffections and a growing burden of non-communicable diseases, the combination of which may lead to disease clustering in ways that are not seen in other regions. The present study explores possibilities for identifying patients with potential comorbidities through factor analysis and relatively simple functional tests. This approach is particularly valuable in the context of a country with limited economic resources and a high prevalence of HIV infection.
The strengths of the study lie in the accuracy of the objective data, which were collected by healthcare professionals, as well as in the interesting combination of these data with subjective information obtained from patient self-reports. Additionally, the accessibility and representativeness of the South African resident population in the study are ensured by providing the respondents with questionnaires translated into three widely spoken languages in the country.
The limitations of the study are also appropriately acknowledged. Among the notable ones are the relatively small sample size and the potential overestimation of disease prevalence, given the recruitment procedure, which involved including patients who were already enrolled in other studies.
The statistical interpretation is carefully conducted, even though the identification of the three clusters may appear somewhat artificial. When exploring associations with functional tests, although causality cannot be inferred, the analysis highlights interactions between multimorbidity and functioning measures that warrant further investigation.
I have only one comment: in Table 3, in the cell for WHODAS-12 Impairment Level / Pattern 1 (HIV-cholesterol-obesity), you wrote "Moderate: (-0.67, -1.46, 0.12)". Please pay attention to the parentheses — the correct format should be: "Moderate: -0.67, (-1.46, 0.12)".
Round 2
Reviewer 1 Report
Comments and Suggestions for Authors
Thank you for addressing all the questions and comments! I do not have any further questions at this point.